# Genetic Mapping of Seven Kinds of Locus for Resistance to Asian Soybean Rust

**DOI:** 10.3390/plants12122263

**Published:** 2023-06-09

**Authors:** Naoki Yamanaka, Luciano N. Aoyagi, Md. Motaher Hossain, Martina B. F. Aoyagi, Yukie Muraki

**Affiliations:** 1Japan International Research Center for Agricultural Sciences (JIRCAS), 1-1 Ohwashi, Tsukuba 305-8686, Japan; 2Department of Plant Pathology, Bangabandhu Sheikh Mujibur Rahman Agricultural University, Salna, Gazipur 1706, Bangladesh

**Keywords:** genetic mapping, *Glycine max*, *Phakopsora pachyrhizi*, resistance gene, *Rpp*, Asian soybean rust

## Abstract

Asian soybean rust (ASR), caused by *Phakopsora pachyrhizi*, is one of the most serious soybean (*Glycine max*) diseases in tropical and subtropical regions. To facilitate the development of resistant varieties using gene pyramiding, DNA markers closely linked to seven resistance genes, namely, *Rpp1*, *Rpp1-b*, *Rpp2*, *Rpp3*, *Rpp4*, *Rpp5*, and *Rpp6*, were identified. Linkage analysis of resistance-related traits and marker genotypes using 13 segregating populations of ASR resistance, including eight previously published by our group and five newly developed populations, identified the resistance loci with markers at intervals of less than 2.0 cM for all seven resistance genes. Inoculation was conducted of the same population with two *P. pachyrhizi* isolates of different virulence, and two resistant varieties, ‘Kinoshita’ and ‘Shiranui,’ previously thought to only harbor *Rpp5*, was found to also harbor *Rpp3*. Markers closely linked to the resistance loci identified in this study will be used for ASR-resistance breeding and the identification of the genes responsible for resistance.

## 1. Introduction

Asian soybean rust (ASR), caused by *Phakopsora pachyrhizi* (Sydow and Sydow), is one of the most serious soybean diseases, especially in major tropical and subtropical soybean (*Glycine max* Merrill)-growing regions worldwide. The huge cost of fungicide control or severe yield loss is common in Asia, Africa, and South America [1,2,3]. Although the introduction of resistant varieties has advantages in terms of both cost and reduced environmental impact, proper selection of resistance genes is essential, considering the virulence of the ASR fungus.

The virulence of ASR varies at the field level, as well as with annual and regional variations. In Mexico, two types of ASR pathogens with largely different virulence characteristics have been reported to be present simultaneously in two regions of the country, approximately 1000 km apart [4]. In Bangladesh, the virulence of ASR fungi changed significantly and increased between 2016 and 2018 [5]. In Uruguay, the variability of ASR pathogens within soybean fields was high [6]. Therefore, ASR-resistant varieties must adapt to these pathogenic variations.

Pyramiding the resistance gene *Rpp*s is an effective means of addressing such diverse and variable ASR virulence. In addition, *Rpp*-pyramiding confers strong synergistic resistance to soybean plants [7,8,9]. The synergistic effects of *Rpp*-pyramiding have been used to develop new ASR-resistant varieties, and their high resistance is effective at the field level [10]. To identify soybean lines that carry multiple *Rpp* genes without DNA selection, multiple *P. pachyrhizi* races are matched to the target genes, and a subsequent progeny test is required to confirm homozygosity/heterozygosity. In addition, it is sometimes difficult to determine the presence of resistance genes by phenotypic values since the effect of *Rpp* gene on the phenotype is influenced by the genetic background, except for *Rpp* gene [11]. Therefore, the selection of resistant soybean lines using DNA markers is essential for the effective and efficient breeding of resistant varieties through *Rpp*-pyramiding. The selection of soybean genotypes by two DNA markers that are closely linked to each other and sandwich a resistance locus contributes not only to variety development but also to the efficient generation of near-isogenic lines (NILs) [12].

In marker-assisted breeding, using DNA markers linked to the target gene, undesirable traits may be introduced along with the target gene owing to linkage drag [13]. Therefore, the DNA markers should be located closer to the target gene. Thus, we reanalyzed previously published linkage maps for the ASR resistance loci *Rpp1*, *Rpp1-b*, *Rpp2*, and *Rpp3* [14,15,16,17], and identified DNA markers that are closely linked to these loci by additive marker positioning. In addition, we analyzed new segregants for *Rpp1-b*, *Rpp4*, *Rpp5*, and *Rpp6* and identified DNA markers closely linked to these loci.

## 2. Results

### 2.1. Genetic Loci of Seven Rpps

*Rpp1* was mapped using three mapping populations—Xiao Jin Huang × BRS184, BRS184 × Himeshirazu, and KS 1034 × BRS184—as described in previous studies [14,15]. Reconstruction of linkage maps by adding new simple sequence repeat (SSR) markers revealed that *Rpp1* was mapped at the position of 10.8 cM from the top of the linkage group between BARCSOYSSR(SSR)18_1793 (5.6 cM) and SSR18_1854 (12.5 cM) in Xiao Jin Huang × BRS184, the position of 11.7 cM between Sct_187 (11.1 cM) and Sat_064 (12.3 cM) in BRS184 × Himeshirazu, and the position of 6.3 cM between Sct_187 (4.3 cM) and Sat_064 (8.7 cM) in KS 1034 × BRS184, respectively (Figure 1, Table 1). The peaks of logarithm of the odds (LOD) scores for numbers of uredinia per lesion (NoU), frequency of lesions with uredinia (%LU), and sporulation level (SL) by additional quantitative trait locus (QTL) analyses were located at the same interval as the SSR markers (Appendix A). As a result, *Rpp1* was found to be located in the genomic region between Sct_187 and SSR18_1854.

*Rpp1-b* was mapped using three mapping populations—BRS184 × PI 594767A, BRS184 × PI 587905, and BRS184 × PI 587855—which were used in previous studies [16,17], and a new mapping population, PI 587880A × No12-1-A. Linkage maps were reconstructed by adding new SSR markers, and *Rpp1-b* was mapped at the position of 16.4 cM from the top of the linkage group between SSR18_1861 (16.4 cM) and Sat_372 (17.0 cM) in BRS184 × PI 594767A, the position of 5.4 cM between SSR18_1861 (5.2 cM) and Sat_372 (5.6 cM) in BRS184 × PI 587905, the position of 13.3 cM between SSR18_1861 (13.1 cM) and SSR18_1862 (13.5 cM) in BRS184 × PI 587855, and the position of 7.4 cM between SSR18_1861 (7.4 cM) and SSR18_1862 (8.4 cM) in BRS184 × PI 587855, respectively (Figure 1, Table 1). *Rpp1-b* locus was considered to be positioned opposite *Rpp1*, across the marker Sat_064 (Figure 1). The peak positions of the LOD values of the QTL analysis for resistance-related traits were also in this vicinity in most cases (Appendix A).

*Rpp2* of Iyodaizu B was mapped as a QTLs for three resistance characters, NoU, %LU, and SL, by a reconstructed linkage map using the Iyodaizu B × BRS184 population (Figure 2). The LOD peaks of QTLs for these three characters were detected at the same position of 0.7 cM-interval between the markers: Satt620 (the position of 8.7 cM from the top of linkage group) and SSR16_0908 (9.4 cM) (Figure 2, Table 1).

*Rpp3* of PI 416,764 was also re-mapped as QTLs for three ASR resistance characters using the BRS184 × PI 416764 population (Figure 3). However, the peak positions of the QTLs for these three characters were different: the position of 5.4 cM for NoU, 2.9 cM for %LU, and 3.6 cM for SL, respectively (Figure 3). Thus, the 3.6 cM interval between the markers SSR06_1466 (position: 2.4 cM) and SSR06_1516 (6.0 cM) containing three peaks of QTLs were selected as the candidate regions where *Rpp3* was located (Table 1, Figure 3).

*Rpp4* of PI 459025 was mapped as a QTL using the BC_3_F_2_ population developed in this study and derived from the BRS184 × PI 459,025 cross (Figure 4). The LOD peaks of three characters are located at the same position of 2.6 cM where three markers—Sc21_3420, SSR18_1557, and SSR18_1568—were located. The 1.0 cM interval between the markers SSR18_1551 (position of 2.1 cM) and SSR18_1572 (3.1 cM) containing the LOD peaks was identified as the candidate region where *Rpp4* was located (Table 1, Figure 4).

*Rpp5* was mapped using two newly developed mapping populations derived from crosses BRS184 × Kinoshita and BRS184 × Shiranui. By the infection of Brazilian BRP-2.5 isolate, in both populations, QTLs for NoU, %LU, and SL were detected in the region of Chromosome 3 where *Rpp5* was previously mapped [18] (Figure 5). However, upon infection with the Japanese E1-4-12 isolate, QTLs for these traits were detected in the region of chromosome 6 where *Rpp3* is located. No QTL was detected in the *Rpp3* region after BRP-2.5 inoculation, and, conversely, no QTL was detected in the *Rpp5* region at E1-4-12 inoculation (Figure 5). *Rpp5* in Kinoshita and Shiranui were determined presenting 1.9 cM interval between the markers, Sat_275 (position: 2.3 cM) and SSR03_0929 (4.2 cM), and 3.3 cM interval between SSR03_0737 (5.2 cM) and SSR03_0929 (8.5 cM), respectively. *Rpp3* in Kinoshita and Shiranui were located at 1.6 cM interval between the markers, Satt460 (position: 0.4 cM) and SSR06_1496 (2.0 cM), and 1.8 cM interval between Sat_263 (1.5 cM) and SSR06_1496 (3.3 cM), respectively. *Rpp5* and *Rpp3* were mapped to the same regions in both populations (Table 1, Figure 5).

*Rpp6* of PI 567102B was mapped as a QTL using the F_2_ population developed in this study and derived from the BRS184 × PI 567102B cross (Figure 6). The LOD peaks of the three characteristics were located at the same position (2.5 cM) from the top of the linkage group. The 2.0 cM interval between the markers SSR18_0356 (1.5 cM) and SSR18_0368 (3.5 cM) containing the LOD peaks was identified as the candidate region where *Rpp6* was located (Table 1, Figure 6).

### 2.2. Genetic Effect of Seven Rpps

For the three traits: NoU, %LU, and SL for resistance, the contribution of each resistance gene to phenotypic variance (variance explained: VE, %) and genetic effects (additive and dominance effects) were calculated based on the genotype of the marker closest to each locus (Table 2, Table 3 and Table 4). For trait NoU, the VE of the resistance genes in each population ranged from 48.99% to 82.60%, with the major genes, *Rpp*s, determining most phenotypes. The gene with the highest VE was *Rpp5* in Kinoshita, and the lowest was *Rpp1-b* in PI 587880A (Table 2). VE in %LU ranged from 0% (not significant) to 72.61%, with the highest value observed for *Rpp5* in Kinoshita (Table 3). No significant QTLs were detected for the %LU trait in the Xiao Jin Huang × BRS184 population (Table 3, Appendix A); therefore, the phenotypic values between genotypes of the nearest marker were not significantly different. The VE for SL ranged from 45.87% to 83.05%, with the highest value for *Rpp5* in Kinoshita and the lowest of *Rpp1* in Xiao Jin Huang (Table 4).

The additive effects in Table 2, Table 3 and Table 4 show the relative effects of one resistant allele relative to the susceptible allele for each resistance gene. Negative values indicate that the resistance allele reduced the number of uredinia, frequency of uredinia-forming lesions, and sporulation levels, with larger absolute values indicating a greater effect of the resistant allele. The biggest additive effects for the three traits NoU, %LU, and SL were observed in *Rpp1-b* of PI 587905 (−1.68), *Rpp1* of KS1034 (−49.57%), and *Rpp1-b* of PI 587905 (−1.48), respectively. However, smaller additive effects for the three traits NoU, %LU, and SL were observed for *Rpp6* of PI 567102B (−0.64), *Rpp1* of Xiao Jin Huang (not significant), and *Rpp1* of Xiao Jin Huang (−0.57). The results showed that the additive effect had a more than two-fold difference in effectiveness, depending on the resistance genes.

Regarding the dominance effect of NoU, all *Rpp*s showed the same negative values as those of the additive effects (Table 2). However, the dominance effect of *Rpp3* in PI 416764 was very small compared to the additive effect, and the dominance effect of *Rpp5* in Kinoshita and Shiranui was negligible. On the other hand, all *Rpp1* or *Rpp1-b*, *Rpp3* in Kinoshita and Shiranui, and *Rpp6* in PI 567102B showed a dominance effect equivalent to the additive effect, with the degree of dominance of resistance alleles being 0.86–1.04. Thus, these genes are considered responsible for the complete dominance of the resistant phenotype. The degree of dominance in *Rpp2* of Iyodaizu B and *Rpp4* of PI 459025 was 0.62 and 0.46, respectively, showing that the resistant phenotype is incomplete dominance. The dominance effect of SL was similar to that of the NoU. The degree of dominance in *Rpp1* of three varieties, *Rpp1-b* of four varieties, *Rpp3* of Kinoshita and Shiranui, and *Rpp6* of PI 567102B ranged from 0.87–1.07, indicating a complete dominance of the resistance phenotype (Table 4). Since the dominance effect in *Rpp3* of PI 416764, and *Rpp5* of Kinoshita and Shiranui was hardly observed (−0.07–0.06), resistant and susceptible phenotypes in these genes were considered codominant. The degrees of dominance in *Rpp2* of Iyodaizu B and *Rpp4* of PI 459025 were 0.55 and 0.24, respectively, showing that the resistant phenotype is incomplete dominance. For %LU, all resistance genes except *Rpp1* of Xiao Jin Huang, *Rpp3* of PI 416764, *Rpp4* of PI 459025, and *Rpp5* of Kinoshita had the same degree of dominance as NoU and SL (Table 3). On the other hand, *Rpp1* of Xiao Jin Huang was not significant for %LU, *Rpp3* of PI 416764 and *Rpp5* of Kinoshita were incompletely dominant for the susceptible phenotype (degree of dominance: −0.68, −0.53), and *Rpp4* of PI 459025 was codominant.

## 3. Discussion

### 3.1. Position of Resistance Loci

Hyten et al. positioned *Rpp1* of PI 200492 between Sat_187 and Sat_064 on chromosome 18 using a population crossing the Williams82 variety [19], and the position of *Rpp1* concluded in this study, between Sct_187 and SSR18_1854, was consistent with this report. Chakraborty et al. mapped the resistance gene of PI 594538A to *Rpp1-b* between Sat_064 and Sat_372 on chromosome 18 [20]. Consistent with the conclusion that *Rpp1-b* is located between SSR18_1861 and SSR18_1862, we restricted the candidate locus to a narrower region.

Yu et al. constructed a detailed genetic map of *Rpp2* in PI 230970 and showed that the gene is located between SSR06_0902 and SSR06_0908 on chromosome 16 [21]. Although this study used a different variety, Iyodaizu B, from PI 230970, the *Rpp2* locus was in the same region as Satt620 (the same as SSR06_0901) and SSR06_0908.

Monteros et al. placed *Rpp3* in Hyuuga between the SSR markers Satt460 and Satt307 on chromosome 6 [22]. Hyten et al. positioned *Rpp3* of PI 462312 between Satt460 and Sat_263 [23]. Aoyagi et al. mapped the resistance genes of the resistant soybean varieties “COL/THAI/1986/THAI-80” and “HM 39” as *Rpp3* close to the SSR marker Satt079 [14]. Because the Satt079 locus is located between Satt460 and Sat_263, the *Rpp3* loci of these four varieties were considered identical. In this study, the peak positions of the LOD scores for the QTL analysis of the three resistance-related traits did not match, so the interval of *Rpp3* presence (3.6 cM between SSR06_1466 and SSR06_1516) in PI 416764 could not be narrowed to a short interval. However, this region of *Rpp3* matched the one identified in the aforementioned four varieties. In addition, the peak LOD score for SL was detected on Satt079, the nearest marker to *Rpp3* (Figure 3). *Rpp3* was located between Satt_460 and SSR06_1496 (1.6 cM) in the Kinoshita population and between Sat_263 and SSR06_1496 (1.8 cM) in the Shiranui population (Figure 5), indicating that these *Rpp3* loci were also located in the same region, as previously reported. These results indicated that there was no discrepancy between the location of *Rpp3* in the four varieties reported in the past and the loci of *Rpp3* identified in this study in the three varieties, which are in very close proximity to the marker Satt079.

Meyer et al. mapped *Rpp4* of PI 459025 to 4.29 cM interval between the SSR markers Sc21-3360 and Satt288 on chromosome 18, using the population derived from the same parental cross population as in this study [24]. The marker closest to *Rpp4* in that report was Satt288, located at a distance of 1.19 cM. In this study, *Rpp4* was placed at a 1 cM interval between SSR18_1551 and SSR18_1572. In the intermediate region between these markers, Sc21_3420, SSR18_1557, and SSR18_1568 shared the same locus, and the LOD scores for the three resistance traits were the highest for these three markers (Figure 4). Sc21_3420, which is the same locus as *Rpp4* in this study, is 4.67 cM away from *Rpp4* on the linkage map of Meyer et al. and the results of the two studies are clearly different. In addition to this marker, we mapped the same markers, Sc21_2716, Sc21_2922, Sc21_4808, and Satt288, as reported by Meyer et al., in this study, however, their positions did not match those of Meyer et al. Since we used the same parents and markers, no speculation can reasonably explain these locus discrepancies. The loci for resistance and DNA markers need to be reconfirmed using another *Rpp4*-segregating population with the same parental varieties.

Lemos et al. (2011) positioned *Rpp5* in Kinoshita approximately 4 cM between the markers Sat_275 and Sat_280 on chromosome 3 by QTL analysis of five resistance-related traits using a population in which *Rpp2*, *Rpp4,* and *Rpp5* segregate [11]. This *Rpp5* position is the same as that of *Rpp5* in PI 200456, PI 471904, and PI 200526 (Shiranui), as reported by Garcia et al. [18], and *Rpp5* in Hyuuga, as reported by Kendrick et al. [25]. We also mapped *Rpp5* in Kinoshita to these regions and narrowed the interval between the markers, where *Rpp5* was present between Sat_275 and SSR03_0929 (Figure 5). In contrast, in Shiranui, the peak position of the LOD score for the three traits was Sat_275; therefore, the interval sandwiching Sat_275 between SSR03_0737 and SSR03_0929 was the interval for the presence of *Rpp5*, and included Kinoshita’s *Rpp5* region (Figure 5). The *Rpp5* loci identified in this study were consistent with those reported previously.

Li et al. located the rust resistance locus on PI 567102B as *Rpp6* between the markers Satt324 and Satt394 on chromosome 18 [26]. In this study, the interval of *Rpp6* was located at 2 cM between the two markers SSR18_0356 and SSR18_0368, and 13.1 cM between these markers (Figure 6). These results showed no discrepancies from previous reports for the six resistance loci other than *Rpp4*.

### 3.2. The Effect of Introducing Resistance Genes

Because soybean is a diploid self-pollinating crop, additive rather than dominance effects appear with the introduction of resistance genes in breeding. The *Rpp1* allele of Xiao Jin Huang was the least effective of all resistance genes analyzed in this study, reducing NoU by 1.38 and SL by 1.14 when introduced as homozygosity (Table 2 and Table 4). No effect was observed on %LU (Table 3). In contrast, *Rpp1* in KS1034 reduced NoU by 1.86, %LU by 99.14, and SL by 2.84. Similarly, there were 1.98-, 2.13-, and 1.68-fold differences in NoU, %LU, and SL between PI 587905, which showed the greatest effect, and the other *Rpp1-b* carrying varieties showed the smallest effect. The difference in effectiveness was more pronounced for different variety alleles at the same locus than among loci, suggesting that the selection of donor varieties will be important at least when *Rpp1* or *Rpp1-b* is used for variety development.

ASR resistance *Rpp* genes are highly race-specific, and the virulence of ASR pathogens in many regions except Japan is diverse [1,4,5,27]. Under such circumstance, *Rpp*-pyramiding can lead to high resistance against *P. pachyrhizi* races that are virulent to the individual *Rpp* genes that have been pyramided [8]. The genetic effect of each resistance gene was revealed in this study. However, to develop useful rust-resistant soybean varieties, it would be desirable to determine the *Rpp* genes to be introduced based on the virulence of the ASR pathogen in the target region and the degree of synergistic effects of *Rpp*-pyramiding.

### 3.3. The Necessity of Marker-Assisted Selection for ASR Resistance and Utility of Markers

All *Rpp1*, *Rpp1-b*, and *Rpp6*, and *Rpp3* of Kinoshita and Shiranui showed complete dominance of the resistance phenotype against the ASR pathogenic strains used in this study. When these resistance genes are introduced into soybean cultivars, it is challenging to distinguish between homozygous and heterozygous lines during selection without DNA markers. Using the DNA markers identified in this study, it was possible to select the presence or absence of resistance genes and homozygosity or heterozygosity in the selected generation without pathogen inoculation and progeny test.

Worldwide, ASR pathogens are diverse in their virulence, and no single resistance gene can guarantee stable resistance. There were highly virulent strains that were pathogenic to all seven resistance genes used in this study, such as the ASR pathogen found in Bangladesh [5]. Additionally, high resistance through the synergic effect of gene pyramiding is useful for developing ASR-resistant varieties [10]. Therefore, gene pyramiding is essential for the development of ASR varieties with a high degree of resistance and stability. For effective and efficient selection of individuals carrying all target genes at homozygosity and synergic effect from a population in which multiple resistance genes are segregated, DNA markers tightly linked to the target genes must be used. This study placed *Rpp1*, *Rpp1-b*, *Rpp2*, *Rpp3*, *Rpp4*, *Rpp5*, and *Rpp6* between the DNA markers of 1.2 cM, 0.4 cM, 0.7 cM, 1.6 cM, 1.0 cM, 1.9 cM, and 2.0 cM in minimum, respectively (Table 1, Figure 1, Figure 2, Figure 3, Figure 4, Figure 5 and Figure 6). Because the probability of double recombination between markers below 2.0 cM is less than 0.04%, we can assume that a resistance gene is present if the two markers flanking the resistance locus show the genotype of the resistant parent. Therefore, these markers can be utilized for marker-assisted breeding for ASR resistance. DNA markers flanking these resistance loci can also be used for positional cloning of resistance genes. In addition, the markers around *Rpp* loci may be used to minimize probability of the introduction of undesirable genes other than *Rpp* by linkage drag when introducing resistance genes by cross-breeding.

### 3.4. Second Resistance Gene Rpp3 in Rpp5-Carrying Varieties, Kinoshita and Shiranui

Shiranui, carrying *Rpp5*, is included in the differential varieties of ASR and has been observed to be more frequently resistant to rust pathogens than other differential varieties with single resistance genes. It was resistant to Bangladeshi ASR samples, with the highest frequency of resistance among the seven genes in the 2016 and 2018 samples [5]. It also showed the highest frequency of resistance to Mexican ASR samples, with resistance in 97.5% of the samples evaluated [4]. It also showed the second-highest frequency of resistance to the South American ASR samples after *Rpp1-b*. In addition, Shiranui showed resistance to the Japanese ASR samples [27]. The resistance tendency of Shiranui may be due to the presence of both *Rpp5* and *Rpp3*. Shiranui (and Kinoshita) showed resistant and immune (resistant) phenotypes to the Brazilian rust strain BRP-2.5, and the Japanese ASR strain E1-4-12 used in this study, respectively; however, only *Rpp5* was resistant to BRP-2.5, and only *Rpp3* was resistant to E1-4-12. In addition to the Japanese varieties Kinoshita and Shiranui, Hyuuga has been reported to possess both *Rpp3* and *Rpp5* [25]. It is not known why ASR-resistant varieties possessing both *Rpp3* and *Rpp5* are frequently found among ASR-resistant Japanese varieties. However, Walker et al. found that the response patterns of rust-resistant soybean varieties to rust were similar in each country of origin [28]. ASR-resistant common ancestor may be present in soybean varieties from southern Japan.

The fact that Kinoshita and Shiranui have *Rpp3* in addition to *Rpp5* and the identification of selection DNA markers for each resistance gene, are important findings of this study for resistance breeding. When these varieties are used as resistance donors, if only the known *Rpp5* is introduced, the new improved varieties may not show resistance to ASR strains, such as E1-4-12, to which Shiranui and Kinoshita are resistant. If Kinoshita and Shiranui are to be used as resistance donors for breeding, both the *Rpp5* and *Rpp3* resistance genes must be efficiently and reliably introduced using markers. In addition, to properly monitor the reactions of soybean resistance genes with the virulence genes of pathogenic races, *Rpp5* and *Rpp3* NILs bred from these varieties must be used instead of Kinoshita and Shiranui. The *Rpp5* and *Rpp3* selection markers identified in this study will be highly effective in developing NILs.

## 4. Materials and Methods

### 4.1. Soybean Populations to Map ASR Resistance Loci

The soybean populations used in this study are listed in Table 5. DNA and phenotypic data from eight previously published mapping populations [14,15,16,17] were also used in this study. In addition to these populations, five new mapping populations were developed, derived from crosses using resistant varieties: PI 587880A, PI 459025, Kinoshita (PI 200487), Shiranui (PI 200526), and PI 567102B, for mapping *Rpp1-b*, *Rpp4*, *Rpp5*, *Rpp5*, and *Rpp6*, respectively. Although the F_2_ mapping population derived from PI 587880A × No12-1-A was developed to obtain *Rpp*-pyramided lines—Py7-1-8-5 carrying *Rpp1-b* from PI 587880A and *Rpp2* and *Rpp5* from No6-12-1A [29]—we used this population to map *Rpp1-b* using the ASR pathogen that is avirulent to *Rpp1-b* but virulent to *Rpp2* and *Rpp5*. Although the parental combination BRS184 × PI 459025 was used for mapping *Rpp4* in a previous study [30], the BC_3_F_2_ mapping population used in this study was developed from an independent cross. Two kinds of *Rpp5*-mapping populations were also newly developed by crossing the ASR-susceptible variety, BRS184, and the known *Rpp5*-carrying resistant varieties, Kinoshita and Shiranui. The resistant variety Kinoshita was crossed with the susceptible variety BRS184 as the germ or pollen parent to obtain F_1_ individuals. An F_2_ mapping population was obtained using the cross BRS184 × PI 567102B while developing a NIL for *Rpp6*. The sizes of the 13 mapping populations are presented in Table 5.

These populations were grown under constant temperature and light conditions in a growth chamber, and the leaves of the plants or leaves detached from the plants were inoculated with *P. pachyrhizi* strains. DNA from the parental varieties and mapping populations was obtained from leaves using the modified CTAB method described in the manual [29].

### 4.2. Resistant Phenotyping in the Mapping Populations

For the analysis of the eight previously published mapping populations [14,15,16,17], the phenotypic data available at the time were used. However, only two resistance traits, the number of uredinia per lesion (NoU) and sporulation level (SL), have been used in previous studies [14,15,16,17]. In this study, the frequency of lesions with uredinia (%LU) was calculated in addition to NoU and SL. For the five mapping populations—PI 587880A × No12-1-A, BRS184 × PI 459025, BRS184 × Kinoshita, BRS184 × Shiranui, and BRS184 × PI 567102B—developed in this study, the first trifoliate were used for inoculation with *P. pachyrhizi* strains listed Table 1 by following the manual [29]. In the case of *Rpp5*-segregating Kinoshita and Shiranui mapping populations, two types of *P. pachyrhizi* strains, BRP-2.5 and E1-4-12, were used for inoculation because preliminary inoculations with these two strains revealed different segregation of resistance in the partial F_2_ populations (data not shown). Each leaflet of the first trifoliate was inoculated with BRP-2.5 and E1-4-12. Inoculation of *P. pachyrhizi* and evaluation of the resistance-related characteristics NoU, %LU, and SL followed the manual [29]. *P. pachyrhizi* strains BRP-2.5, T1-2, and E1-4-12 used in this study were obtained from our previous studies [7,8,27]. Urediniospores of these *P. pachyrhizi* strains were inoculated at a concentration of 3.6–5.8 × 10^4^ urediniospores/mL.

### 4.3. Genotyping of DNA Markers on the Mapping Populations

For the eight mapping populations—Xiao Jin Huang × BRS184, BRS184 × Himeshirazu, KS 1034 × BRS184, BRS184 × PI 594767A, BRS184 × PI 587905, BRS184 × PI 587855, Iyodaizu B × BRS184, and BRS184 × PI 416764—previously published [14,15,16,17], additional SSR markers were newly analyzed in this study using the DNA solution preserved in −20 °C, respectively. For the five mapping populations—PI 587880A × No12-1-A, BRS184 × PI 459025, BRS184 × Kinoshita, BRS184 × Shiranui, and BRS184 × PI 567102B—the newly developed SSR markers around the target *Rpp* regions were chosen for analysis. For the PI 587880A × No12-1-A population, SSR markers around the target *Rpp1-b* locus [16], SSR markers around *Rpp2* and *Rpp5* [18,30] were also analyzed to confirm that these two loci did not contribute to the resistance phenotype in this mapping population for the *P. pachyrhizi* strain BRP-2.1. Regarding the mapping populations, BRS184 × PI 459025 for *Rpp4* and BRS184 × PI 567102B for *Rpp6* SSR markers around these two regions [26,30] were chosen and used for mapping.

In the case of two mapping populations—BRS184 × Kinoshita and BRS184 × Shiranui for *Rpp5*—a small subset (22 and 24 F_2_ plants, respectively) of F_2_ plants independent of the main mapping populations (130 and 138 F_2_ plants, respectively; Table 5) were evaluated for their resistance phenotypes against E1-4-12 and genotyped using polymorphic and codominant SSR markers Sat_372, Satt380, Sat_263, Satt288, Sat_280, and Satt324 linked to *Rpp1* (*Rpp1-b*)*, 2*, *3*, *4*, *5* and *6*, respectively. A significant association between NoU and marker genotypes was found in the E1-4-12 and *Rpp3* markers, and fully mapped populations were analyzed by SSR markers around the regions of *Rpp3* and *Rpp5*. The protocols for the PCR and genotyping of SSR markers are described in the manual [29].

### 4.4. Linkage Analysis of SSR Markers and Deciding Rpp Loci

MAPMAKER/EXP 3.0b [31] was used to determine the order of the SSR marker loci in the mapping populations. The Kosambi mapping function was used to convert the recombination values into map distances (cM). The minimum logarithm of the odds (LOD) score and the maximum genetic distance for linkage map construction were 3.0 and 37.2 cM, respectively. For the three mapping populations for *Rpp1* and four mapping populations for *Rpp1-b*, F_2_ plants were classified as resistant or susceptible by mapping resistance loci as a single resistance factor together with SSR markers, according to our previous studies [14,15,16,17].

In the other mapping populations for *Rpp2-Rpp6*, genomic regions significantly associated with NoU, %LU, and SL were determined using interval mapping (IM) and composite interval mapping (CIM) from the Windows QTL Cartographer v.2.5 [32]. Other parameters defined for the QTL analysis were 0.5 cM walk speed, 1000 permutations (permutation test), and a 0.01 significance level, following the methodology and parameters employed by [15]. For the CIM analysis, a standard model with five control markers, a window size of 10 cM, and forward- and backward-regression models were applied together with the same parameters as the IM in this study. The resistance loci identified by QTL analysis were determined in the marker interval where the maximum LOD scores for all three characteristics, NoU, %LU, and SL, were detected.

Phenotypic data of resistance and genotypic data of DNA markers for all 13 segregation populations used for mapping can be found in Appendix A.

## Figures and Tables

**Figure 1 plants-12-02263-f001:**
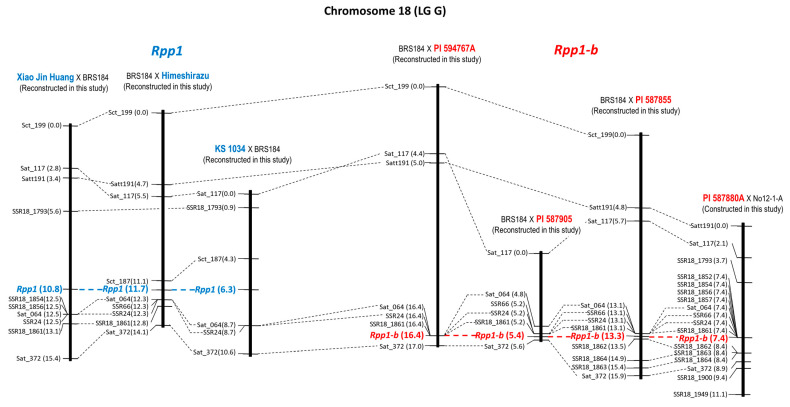
Genetic linkage map of chromosomes (Chr.) 18 (linkage group: LG G) where *Rpp1* and *Rpp1-b* are located. The linkage maps, except for PI 587880A × No12-1-A, have been reconstructed by including new markers into the maps previously reported [14,15,16,17]. Marker name, distance from the top of the linkage group, and *Rpp* loci are shown to the left of each linkage group. *Rpp1* locus and the varieties carrying *Rpp1* are shown in blue, and *Rpp1-b* locus and the varieties carrying *Rpp1-b* are shown in red, respectively.

**Figure 2 plants-12-02263-f002:**
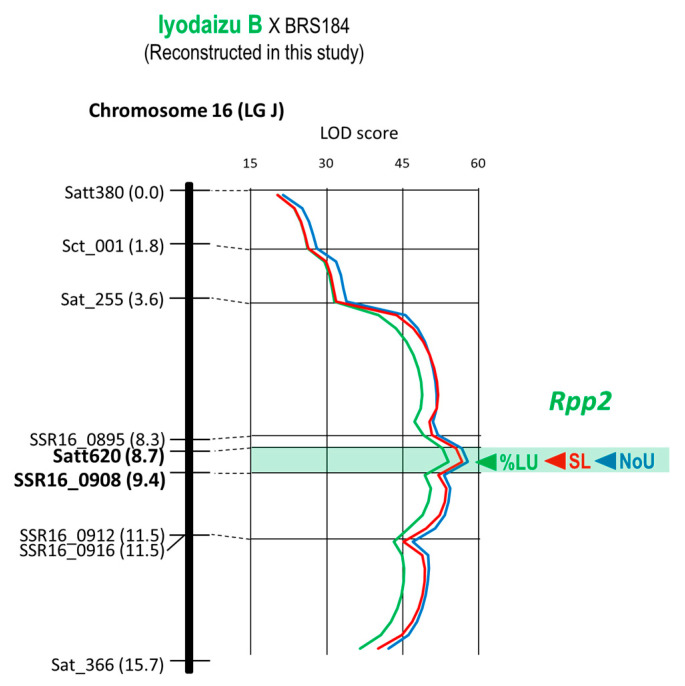
Genetic linkage map of Chr. 16 (LG J) where *Rpp2* is located. This map has been reconstructed by including new markers into the map previously reported [15] and reanalyzing QTLs. Marker name, distance from the top of the linkage group, and *Rpp* name are shown to the left of each linkage group. At the right of the linkage group, LOD curves/peaks of QTLs for NoU and SL are also shown. Estimated region of the *Rpp2* locus flanked by DNA markers is colored green.

**Figure 3 plants-12-02263-f003:**
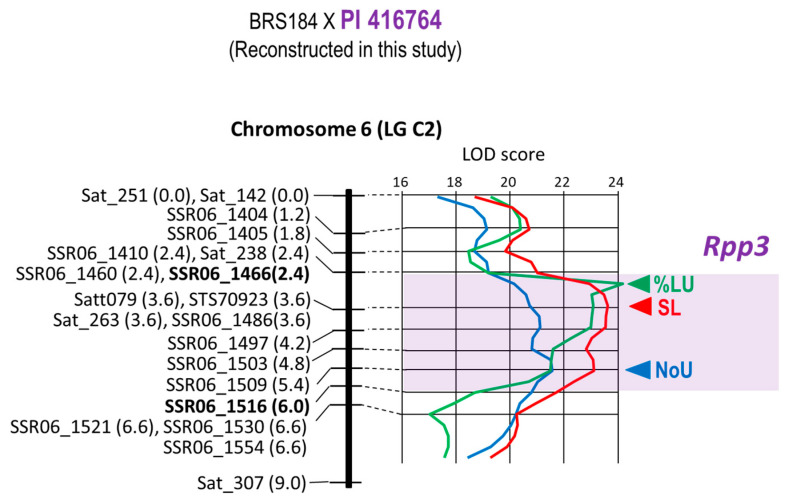
Genetic linkage map of Chromosome 6 (LG C2) where *Rpp3* is located. This map has been reconstructed by including new markers into the map previously reported [16] and reanalyzing QTLs. Name of the marker and distance from the top of the linkage group are shown to the left of linkage group. At the right of the linkage group, LOD curves of QTLs for NoU, %LU, and SL together with the peaks (arrowhead) of LOD curves are shown. Estimated region of the *Rpp3* locus flanked by DNA markers is colored purple.

**Figure 4 plants-12-02263-f004:**
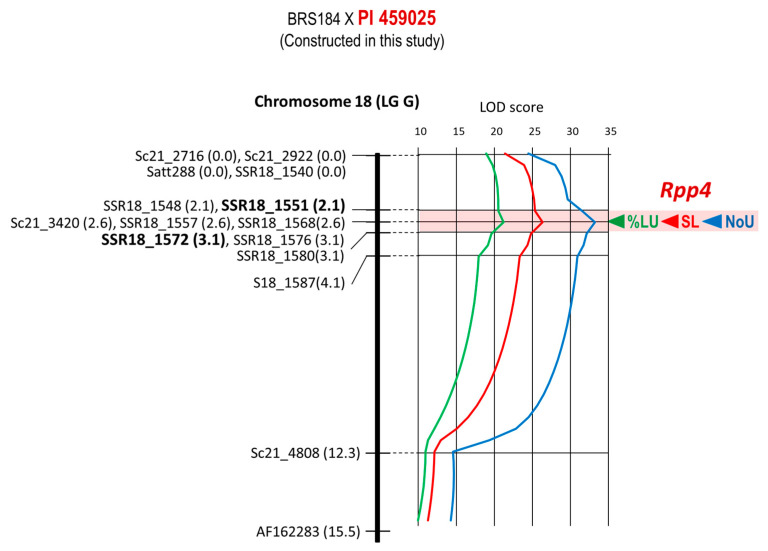
Genetic linkage map of Chromosome 18 (LG G) where *Rpp4* is located. Name of marker and distance from the top of the linkage group are shown to the left of linkage group. At the right of the linkage group, LOD curves of QTLs for NoU, %LU, and SL together with the peaks (arrowhead) of LOD curves are also shown. Estimated region of the *Rpp4* locus flanked by DNA markers is colored red.

**Figure 5 plants-12-02263-f005:**
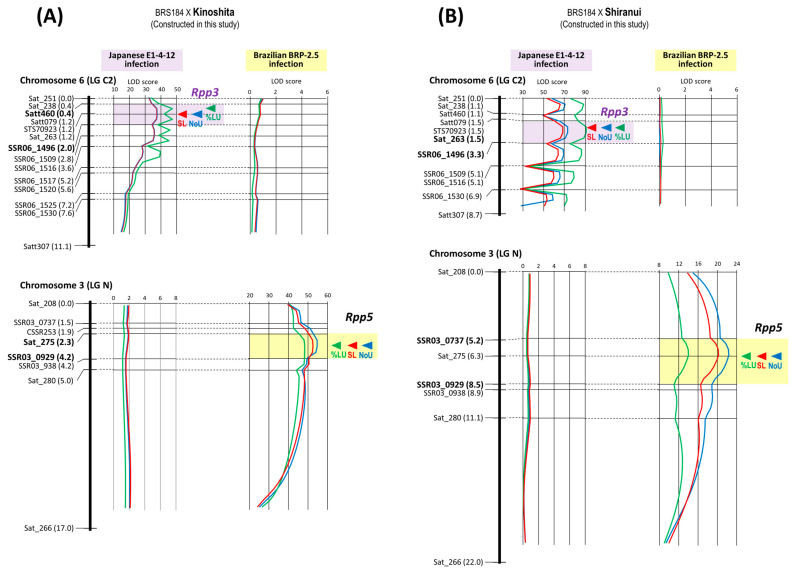
Genetic linkage maps of Chr. 6 (LG N) and Chr. 3 (LG N) using the F_2_ populations derived from the cross between BRS184 and Kinoshita (**A**) and the cross between BRS184 and Shiranui (**B**), where *Rpp3* and *Rpp5* are, respectively, located. Marker name and distance from the top of the linkage group are shown to the left of linkage group. To the right of linkage group, LOD curves and peaks of QTLs for NoU, %LU, and SL against Japanese E1-4-12 and Brazilian BRP-2.5 are shown, respectively. Estimated regions of the *Rpp3* and *Rpp5* loci flanked by DNA markers are colored purple and yellow, respectively.

**Figure 6 plants-12-02263-f006:**
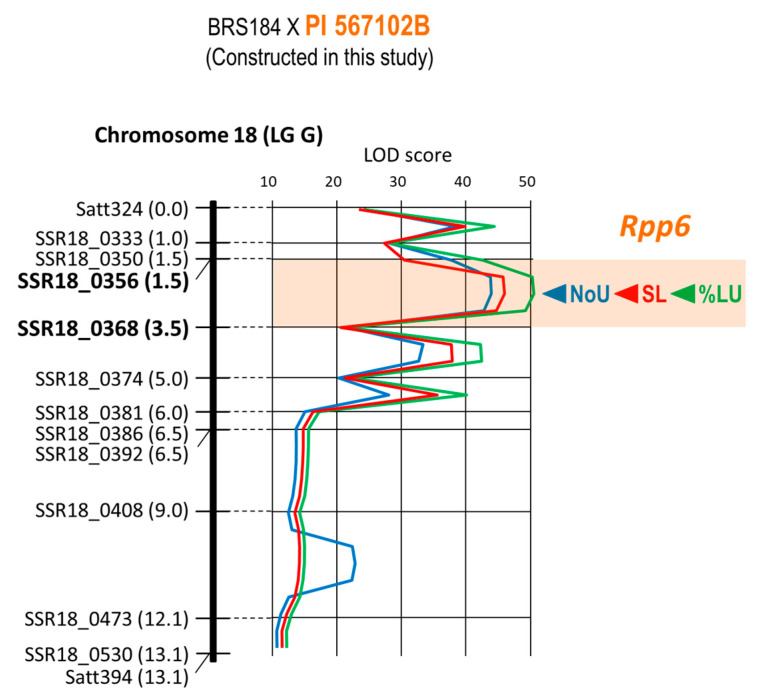
Genetic linkage map of Chromosome 18 (LG G) where *Rpp6* is located. Name of marker and distance from the top of the linkage group are shown to the left of linkage group. At the right of the linkage group, LOD curves of QTLs for NoU, %LU, and SL together with the peaks (arrowhead) of LOD curves are also shown. Estimated region of the *Rpp6* locus flanked by DNA markers is colored orange.

**Table 1 plants-12-02263-t001:** Genetic maps of resistance genes to Asian soybean rust (ASR) constructed using DNA markers.

Pop.	Parental Cross	Target Gene	Chromo-some	Numbers of Markers	Numbers of Marker Loci	Total Length of Linkage Group	Position of Target *Rpp*s ^2^	One of the Nearest Markers ^3^
1	Xiao Jin Huang × BRS184	*Rpp1*	Chr. 18	10	7	15.4 cM	10.8 cM	SSR18_1854
2	BRS184 × Himeshirazu	*Rpp1*	Chr. 18	9	7	14.1 cM	11.7 cM	Sct_187
3	KS 1034 × BRS184	*Rpp1*	Chr. 18	6	5	10.6 cM	6.3 cM	Sct_187
4	BRS184 × PI 594767A	*Rpp1-b*	Chr. 18	7	5	17.0 cM	16.4 cM	SSR18_1861
5	BRS184 × PI 587905	*Rpp1-b*	Chr. 18	6	4	5.6 cM	5.4 cM	SSR18_1861
6	BRS184 × PI 587855	*Rpp1-b*	Chr. 18	11	8	15.9 cM	13.3 cM	SSR18_1861
7	PI 587880A × No12-1-A ^1^	*Rpp1-b*	Chr. 18	17	8	11.1 cM	7.4 cM	SSR18_1861
8	Iyodaizu B × BRS184	*Rpp2*	Chr. 16	9	8	15.7 cM	8.7–9.4 cM	SSR16_0908
9	BRS184 × PI 416764	*Rpp3*	Chr. 6	20	11	9.0 cM	2.4–6.0 cM	Satt079 ^4^
10	BRS184 × PI 459025	*Rpp4*	Chr. 18	15	7	15.5 cM	2.1–3.1 cM	SSR18_1568
11	BRS184 × Kinoshita	*Rpp5*	Chr. 3	8	8	17.0 cM	2.3–4.2 cM	SSR03_0929
		*Rpp3*	Chr. 6	14	11	11.1 cM	0.4–2.0 cM	Sat_263
12	BRS184 × Shiranui	*Rpp5*	Chr. 3	7	7	22.0 cM	5.2–8.5 cM	Sat_275
		*Rpp3*	Chr. 6	11	7	8.7 cM	1.5–3.3 cM	Sat_263
13	BRS184 × PI 567102B	*Rpp6*	Chr. 18	13	10	13.1 cM	1.5–3.5 cM	SSR18_0356

^1^ *Rpp2* and *Rpp5* regions are not included in this table because the pathogen is virulent to these genes. ^2^ Positions of *Rpp* loci identified as quantitative trait loci (QTLs) are shown as intervals. ^3^ For QTL, the marker closest to the peak LOD value. Other markers may be present and co-located with the listed marker or at the same distance from *Rpp*. ^4^ The marker closest to the SL peak within the interval was considered representative because the peaks of the LOD values for the three traits did not match.

**Table 2 plants-12-02263-t002:** Genetic effects and variance explained of resistance genes for the character trait, numbers of uredinia per lesion (NoU). Data in this table are based on the nearest marker for each resistance gene.

Pop.	Parental Cross	Target Gene	The Nearest Markers	Pathogen	Additive Effect ^1^	Dominance Effect ^1^	Degree of Dominance	Variance Explained (%)
1	Xiao Jin Huang × BRS184	*Rpp1*	SSR18_1854	T1-2	−0.69	−0.72	1.04	48.99%
2	BRS184 × Himeshirazu	*Rpp1*	Sct_187	E1-4-12	−0.76	−0.76	1.00	64.65%
3	KS 1034 × BRS184	*Rpp1*	Sct_187	E1-4-12	−0.93	−0.80	0.86	60.54%
4	BRS184 × PI 594767A	*Rpp1-b*	SSR18_1861	T1-2	−1.40	−1.40	1.00	55.93%
5	BRS184 × PI 587905	*Rpp1-b*	SSR18_1861	T1-2	−1.68	−1.55	0.92	64.18%
6	BRS184 × PI 587855	*Rpp1-b*	SSR18_1861	E1-4-12	−0.85	−0.85	1.00	65.41%
7	PI 587880A × No12-1-A	*Rpp1-b*	SSR18_1861	BRP-2.1	−0.86	−0.87	1.01	48.94%
8	Iyodaizu B × BRS184	*Rpp2*	SSR16_0908	E1-4-12	−0.81	−0.51	0.62	69.42%
9	BRS184 × PI 416764	*Rpp3*	Satt079	T1-2	−1.05	−0.22	0.21	65.73%
10	BRS184 × PI 459025	*Rpp4*	SSR18_1568	E1-4-12	−0.90	−0.41	0.46	70.59%
11	BRS184 × Kinoshita	*Rpp5*	SSR03_0929	BRP-2.5	−1.15	−0.03	0.02	82.60%
		*Rpp3*	Sat_263	E1-4-12	−0.81	−0.74	0.91	51.50%
12	BRS184 × Shiranui	*Rpp5*	Sat_275	BRP-2.5	−0.97	−0.09	0.09	52.29%
		*Rpp3*	Sat_263	E1-4-12	−0.82	−0.81	0.99	52.37%
13	BRS184 × PI 567102B	*Rpp6*	SSR18_0356	E1-4-12	−0.64	−0.61	0.96	59.07%

^1^ Genetic effects of the resistant allele contrasted with the susceptible allele of BRS184 in all populations, except population 7. In population 7, the susceptible allele of *Rpp1-b* was the No12-1-A allele.

**Table 3 plants-12-02263-t003:** Genetic effects and variance explained of resistance genes for the character trait, frequency of lesions with uredinia (%LU). Data in this table are based on the nearest marker for each resistance gene.

Pop.	Parental Cross	Target Gene	The Nearest Markers	Pathogen	Additive Effect ^1^	Dominance Effect ^1^	Degree of Dominance	Variance Explained (%)
1	Xiao Jin Huang × BRS184	*Rpp1*	SSR18_1854	T1-2	ns	ns	ns	ns
2	BRS184 × Himeshirazu	*Rpp1*	Sct_187	E1-4-12	−46.17%	−46.17%	1.00	66.58%
3	KS 1034 × BRS184	*Rpp1*	Sct_187	E1-4-12	−49.57%	−40.84%	0.82	59.62%
4	BRS184 × PI 594767A	*Rpp1-b*	SSR18_1861	T1-2	−23.21%	−23.06%	0.99	57.61%
5	BRS184 × PI 587905	*Rpp1-b*	SSR18_1861	T1-2	−49.47%	−37.37%	0.76	69.03%
6	BRS184 × PI 587855	*Rpp1-b*	SSR18_1861	E1-4-12	−47.62%	−47.42%	1.00	67.78%
7	PI 587880A × No12-1-A	*Rpp1-b*	SSR18_1861	BRP-2.1	−44.71%	−45.24%	1.01	62.11%
8	Iyodaizu B × BRS184	*Rpp2*	SSR16_0908	E1-4-12	−45.72%	−21.04%	0.46	72.25%
9	BRS184 × PI 416764	*Rpp3*	Satt079	T1-2	−33.63%	22.76%	−0.68	59.92%
10	BRS184 × PI 459025	*Rpp4*	SSR18_1568	E1-4-12	−38.64%	−1.12%	0.03	63.43%
11	BRS184 × Kinoshita	*Rpp5*	SSR03_0929	BRP-2.5	−40.19%	21.15%	−0.53	72.61%
		*Rpp3*	Sat_263	E1-4-12	−42.26%	−36.57%	0.87	55.69%
12	BRS184 × Shiranui	*Rpp5*	Sat_275	BRP-2.5	−29.98%	5.70%	−0.19	36.61%
		*Rpp3*	Sat_263	E1-4-12	−42.50%	−41.42%	0.97	55.93%
13	BRS184 × PI 567102B	*Rpp6*	SSR18_0356	E1-4-12	−40.00%	−38.23%	0.96	59.38%

^1^ Genetic effects of the resistant allele contrasted with the susceptible allele of BRS184 in all populations, except population 7. In population 7, the susceptible allele of *Rpp1-b* was the No12-1-A allele.

**Table 4 plants-12-02263-t004:** Genetic effects and variance explained of resistance genes for the character trait, sporulation level (SL). Data in this table are based on the nearest marker for each resistance gene.

Pop.	Parental Cross	Target Gene	The Nearest Markers	Pathogen	Additive Effect ^1^	Dominance Effect ^1^	Degree of Dominance	Variance Explained (%)
1	Xiao Jin Huang × BRS184	*Rpp1*	SSR18_1854	T1-2	−0.57	−0.61	1.07	45.87%
2	BRS184 × Himeshirazu	*Rpp1*	Sct_187	E1-4-12	−0.94	−0.94	1.00	66.32%
3	KS 1034 × BRS184	*Rpp1*	Sct_187	E1-4-12	−1.42	−1.23	0.87	60.54%
4	BRS184 × PI 594767A	*Rpp1-b*	SSR18_1861	T1-2	−1.46	−1.45	1.00	57.38%
5	BRS184 × PI 587905	*Rpp1-b*	SSR18_1861	T1-2	−1.48	−1.33	0.90	66.92%
6	BRS184 × PI 587855	*Rpp1-b*	SSR18_1861	E1-4-12	−1.00	−1.00	1.00	66.84%
7	PI 587880A × No12-1-A	*Rpp1-b*	SSR18_1861	BRP-2.1	−0.88	−0.90	1.01	59.73%
8	Iyodaizu B × BRS184	*Rpp2*	SSR16_0908	E1-4-12	−0.96	−0.52	0.55	71.16%
9	BRS184 × PI 416764	*Rpp3*	Satt079	T1-2	−1.03	0.04	−0.04	71.63%
10	BRS184 × PI 459025	*Rpp4*	SSR18_1568	E1-4-12	−0.99	−0.24	0.24	69.12%
11	BRS184 × Kinoshita	*Rpp5*	SSR03_0929	BRP-2.5	−1.10	0.08	−0.07	83.05%
		*Rpp3*	Sat_263	E1-4-12	−0.98	−0.88	0.90	51.71%
12	BRS184 × Shiranui	*Rpp5*	Sat_275	BRP-2.5	−1.04	−0.07	0.06	48.93%
		*Rpp3*	Sat_263	E1-4-12	−1.09	−1.07	0.99	52.55%
13	BRS184 × PI 567102B	*Rpp6*	SSR18_0356	E1-4-12	−0.80	−0.77	0.96	58.38%

^1^ Genetic effects of the resistant allele contrasted with the susceptible allele of BRS184 in all populations, except population 7. In population 7, the susceptible allele of *Rpp1-b* was the No12-1-A allele.

**Table 5 plants-12-02263-t005:** Information on soybean populations and Japanese and Brazilian inocula for mapping *Rpp* resistance loci against Asian soybean rust.

Pop.	Target Gene	Parental Cross	Population Structure	Numbers of Genotypes	Inoculation Target ^3^	Pathogenic Strain	Reference
1	*Rpp1*	Xiao Jin Huang × BRS184	F_2_	90	Plant	T1-2	[15]
2	*Rpp1*	BRS184 × Himeshirazu	F_2_	120	Plant	E1-4-12	[15]
3	*Rpp1*	KS 1034 × BRS184	F_2_	163	Leaf	E1-4-12	[14]
4	*Rpp1-b*	BRS184 × PI 594767A	F_2_	82	Plant	T1-2	[16]
5	*Rpp1-b*	BRS184 × PI 587905	F_2_	117	Plant	T1-2	[16]
6	*Rpp1-b*	BRS184 × PI 587855	F_2_	106	Plant	E1-4-12	[17]
7	*Rpp1-b* ^1^	PI 587880A × No12-1-A	F_2_	96	Plant	BRP-2.1	-
8	*Rpp2*	Iyodaizu B × BRS184	F_2_	143	Plant	E1-4-12	[15]
9	*Rpp3*	BRS184 × PI 416764	F_2_	86	Plant	T1-2	[16]
10	*Rpp4*	BRS184 × PI 459025	BC_3_F_2_	97	Plant	E1-4-12	-
11	*Rpp5,* (*Rpp3*)	BRS184 × Kinoshita ^2^	F_2_	130	Leaf	BRP-2.5, E1-4-12	-
12	*Rpp5,* (*Rpp3*)	BRS184 × Shiranui	F_2_	138	Leaf	BRP-2.5, E1-4-12	-
13	*Rpp6*	BRS184 × PI 567102B	F_2_	100	Plant	E1-4-12	-

^1^ *Rpp2* and *Rpp5* except for *Rpp1-b* are also segregated in this population since No6-12-A carries these genes. ^2^ To develop this population, the resistant variety Kinoshita was crossed with the susceptible variety BRS184 as the germ or pollen parent to obtain F_1_ individuals. ^3^ Plant: pathogen was inoculated on the leaves of the plants and the disease was incubated on the plants until evaluation; leaf: pathogen was inoculated on leaves detached from the plants and the disease was cultured on leaves until evaluation.

## Data Availability

Data are contained within the article or Appendix A.

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
