# Peer review of "Genetic Mapping of Seven Kinds of Locus for Resistance to Asian Soybean Rust"

_plants, 2023, doi:10.3390/plants12122263_

Round 1

Reviewer 1 Report

This manuscript discusses Asian soybean rust (ASR), a destructive disease affecting soybean crops in tropical and subtropical regions. The researchers aimed to develop resistant soybean varieties by combining multiple genes for resistance. By analyzing the linkage between resistance traits and marker genotypes in 13 populations, including both previously published and newly developed populations, they successfully pinpointed the loci associated with all seven resistance genes. The article is effectively written and presents a wealth of valuable results. The identification of markers in this study holds great significance for the development of ASR-resistant soybean varieties and advancing our understanding of the genes underlying resistance. Furthermore, the comprehensive discussion section, which incorporates a wide range of relevant articles, demonstrates its readiness for publication.

Author Response

Thank you for your evaluation and comments on our paper. Another reviewer suggested some revisions and we have made some changes, but the revised manuscript is unchanged from the original in terms of argument and organization.

Reviewer 2 Report

The article “Genetic mapping of seven kinds of locus for resistance to Asian soybean rust”, of Naoki Yamanaka, Luciano N. Aoyagi, Md. Motaher Hossain, Martina B. F. Aoyagi and Yukie Muraki is devoted to important and actual topic. Indeed, soybean rust caused by Phakopsora pachyrhizi is one of the most important foliar diseases of soybean [Glycine max (L.) Merr.] in tropical and subtropical areas. Although seven Rpp resistance gene loci have been reported, extensive pathotype variation in and among fungal populations increases the importance of identifying additional information about genes and loci associated with rust resistance. Proper selection of resistance genes is essential, considering high virulence of the ASR fungus, given that ASR virulence varies at the field level and also has annual and regional variations. For this purpose the synergistic effects of Rpp-pyramiding have been used effectively to develop new ASR-resistant at the field level varieties.

The authors emphasise that, in marker-assisted breeding, using DNA markers linked to the target gene, undesirable traits may be introduced along with the target gene owing to linkage drag. Correspondingly, the DNA markers should be located closer to the target gene. Therefore, the authors attempted to improve on their earlier findings, reanalyzing previously published linkage maps for the ASR resistance loci Rpp1, Rpp1-b, Rpp2, and Rpp3, and identified DNA markers that are closely linked to these loci by additive marker positioning. In addition, they analyzed new segregants for Rpp1-b, Rpp4, Rpp5, and Rpp6 and identified DNA markers closely linked to these loci.

An important continuation of earlier work of this research group is the addition of five newly developed populations identified the resistance loci with markers at intervals of less than 2.0 cM for all seven resistance genes. It is worth highlighting the part of the authors' work on the joint presence of Rpp3 in Rpp5-carrying Japanese varieties Kinoshita and Shiranui to the differential varieties of ASR. It was observed significant finding, that they are to be more frequently resistant to rust pathogens than other differential varieties with single resistance genes. Potentially all this information about genomic regions associated with rust resistance will further efforts to develop soybean cultivars with broad and durable resistance to soybean rust.

Some suggestions:

(73) “LOD scores” is given without abbreviation explanation.

(20, 60, 385, 418, 438) Authors use the sentence “previously published by our group” in Abstract (20), “previously published” once in Introduction (60) and three times in Materials and Methods (385, 418, 438). Only once in Materials and Methods this sentence followed by reference of work of your research group [15–17]. Probably it is better to refer to work of your group in all places except Abstract.

(50-51) “In addition, the effect of Rpp gene on the phenotype is influenced by the genetic background except for Rpp gene [11]”.  The phrase is not clear/

(279-291) In the discussion of the article, attention is drawn to the difference noted in the locus discrepancies described for the Rpp4 with the linkage map of Meyer et al. [24] and the results of the present study. They are clearly different, despite the fact that the authors of both groups used the same parents and markers. This contradiction needs to be explained somehow in the following research.  

(318-319) The authors suggested that “the selection of donor varieties may influence the effectiveness of the introduction of resistance genes, especially Rpp1 and Rpp1-b, in variety development.”   Why it stands out in particular Rpp1 and Rpp1-b?

(323-327) “Therefore, to develop useful rust-resistant soybean varieties, it would be desirable to determine the Rpp genes to be introduced based on the virulence of the ASR pathogen in the target region and the degree of synergistic effects of Rpp-pyramiding rather than on differences in the degree of additive effects of the resistance genes detected in this study.” The conclusion is quite logical, but it is not entirely clear how it is supported by the work.

(342-344) “For effectively and efficiently selection of individuals homozygous for all target genes from a population in which multiple resistance genes are segregated, DNA markers tightly linked to the target genes must be used.”   - and perhaps consider the effect of their synergies???

(351-353) “In addition, the markers around Rpp loci may be used to minimize the introduction of undesirable genes other than Rpp by linkage drag when introducing resistance genes by crossbreeding.”  Perhaps is better to say, that minimize probability of the introduction.

(368-370) Authors are concluded, that “ASR- resistant varieties possessing both Rpp3 and Rpp5 are frequently found among ASR-resistant varieties of Japanese origin”, based only on three Japanese varieties Kinoshita, Shiranui and Hyuuga. Perhaps, it will be interesting represent this fact as phylogenetic three or give at least more references to confirm this information. So, it will be clearer how they are related. (367-368) “There is no evidence that these Japanese varieties are closely related”. (see for example: “Dendrogram depicting the genetic relationship among soybean plant introductions in a panel of germplasm accessions evaluated for their reactions to soybean rust”. Walker et al., 2022 doi:10.1007/s00122-022-04168-y).

Author Response

> Thank you for your evaluation and comments on our paper. We have addressed each of your individual suggestions for revision as follows. We believe you will appreciate that the current manuscript is now possible to be published.

Some suggestions:

(73) “LOD scores” is given without abbreviation explanation.

> Spelled out "LOD".

(20, 60, 385, 418, 438) Authors use the sentence “previously published by our group” in Abstract (20), “previously published” once in Introduction (60) and three times in Materials and Methods (385, 418, 438). Only once in Materials and Methods this sentence followed by reference of work of your research group [15–17]. Probably it is better to refer to work of your group in all places except Abstract.

>In all sections except the abstract, "previous studies" have been followed by citations.

(50-51) “In addition, the effect of Rpp gene on the phenotype is influenced by the genetic background except for Rpp gene [11]”.  The phrase is not clear/

> As you pointed out, the meaning is not clear as it is, so I added "it is sometimes difficult to determine the presence of resistance genes by phenotypic values since" before this sentence.

(279-291) In the discussion of the article, attention is drawn to the difference noted in the locus discrepancies described for the Rpp4 with the linkage map of Meyer et al. [24] and the results of the present study. They are clearly different, despite the fact that the authors of both groups used the same parents and markers. This contradiction needs to be explained somehow in the following research.  

> I agree with your idea. Thus, “The loci for resistance and DNA markers need to be reconfirmed using another Rpp4-segregating population with the same parental varieties.” was added in this paragraph.

(318-319) The authors suggested that “the selection of donor varieties may influence the effectiveness of the introduction of resistance genes, especially Rpp1 and Rpp1-b, in variety development.”   Why it stands out in particular Rpp1 and Rpp1-b?

> The importance of donor selection for all but Rpp1 and Rpp1-b was not demonstrated in this study. Therefore, the wording was changed as follows.

“….suggesting that the selection of donor varieties will be important at least when Rpp1 or Rpp1-b is used for variety development.”

(323-327) “Therefore, to develop useful rust-resistant soybean varieties, it would be desirable to determine the Rpp genes to be introduced based on the virulence of the ASR pathogen in the target region and the degree of synergistic effects of Rpp-pyramiding rather than on differences in the degree of additive effects of the resistance genes detected in this study.” The conclusion is quite logical, but it is not entirely clear how it is supported by the work.

> This conclusion is based on the citations listed. The wording has been changed so as not to mislead the reader into thinking that it was derived from the findings of this study.

(342-344) “For effectively and efficiently selection of individuals homozygous for all target genes from a population in which multiple resistance genes are segregated, DNA markers tightly linked to the target genes must be used.”   - and perhaps consider the effect of their synergies???

> Yes. The individuals to be selected are those with multiple genes and their synergistic effects. The sentence has been revised.

(351-353) “In addition, the markers around Rpp loci may be used to minimize the introduction of undesirable genes other than Rpp by linkage drag when introducing resistance genes by crossbreeding.”  Perhaps is better to say, that minimize probability of the introduction.

> I have revised it as you suggested.

(368-370) Authors are concluded, that “ASR- resistant varieties possessing both Rpp3 and Rpp5 are frequently found among ASR-resistant varieties of Japanese origin”, based only on three Japanese varieties Kinoshita, Shiranui and Hyuuga. Perhaps, it will be interesting represent this fact as phylogenetic three or give at least more references to confirm this information. So, it will be clearer how they are related. (367-368) “There is no evidence that these Japanese varieties are closely related”. (see for example: “Dendrogram depicting the genetic relationship among soybean plant introductions in a panel of germplasm accessions evaluated for their reactions to soybean rust”. Walker et al., 2022 doi:10.1007/s00122-022-04168-y)

> I have replaced the part you indicated with the following text, using the reference you provided as a citation. “It is not known why ASR-resistant varieties possessing both Rpp3 and Rpp5 are frequently found among ASR-resistant Japanese varieties. However, Walker et al. found that the response patterns of rust-resistant soybean varieties to rust were similar in each country of origin [28]. ASR-resistant common ancestor may be present in soybean varieties from southern Japan.”